# Simulation of Epitaxial Film–Substrate Interaction Potential

**Sergey V. Belim [1,\*], Ilya V. Tikhomirov [1] and Igor V. Bychkov [2]**

1. Department of Physics, Omsk State Technical University, 644050 Omsk, Russia; ivtikhomirov@omgtu.ru
2. Department of Radiophysics and Electronics, Chelyabinsk State University, 454001 Chelyabinsk, Russia; bychkov@csu.ru
* Correspondence: sbelim@mail.ru

**Abstract:** The formation of the substrate surface potential based on the Lennard-Jones two-particle potential is investigated in this paper. A simple atom's square lattice on the substrate surface is considered. The periodic potential of the substrate atoms is decomposed into a Fourier series. The amplitude ratio for different frequencies has been examined numerically. The substrate potential is approximated with high accuracy by the Frenkel–Kontorova potential at most parameter values. There is a field of parameters in which the term plays a significant role, with a period half as long as the period of the substrate atoms. The ground state of the monoatomic film is modeled on the substrate potential. The film may be in both crystalline and amorphous phases. The transition to the amorphous phase is associated with a change in the landscape of the substrate potential. There are introduced order parameters for structural phase transition in the thin film. When changing the parameters of the substrate, the order parameter experiences a jump when changing the phase of the film.

**Keywords:** surface potential; Lennard-Jones two-particle potential; Frenkel–Kontorova potential; thin film; structural phase transition





## 1. Introduction

The substrate potential determines the basic properties of epitaxial films. The film's crystal lattice is formed in the sputtering step under the influence of the substrate. The relative arrangement of the film atoms is determined by the substrate potential. The mechanical, thermal, magnetic, and optical properties of the epitaxial film can be altered by acting on the substrate. Changing the size or symmetry of a substrate crystal lattice changes its surface potential. The calculation of the substrate surface potential makes it possible to select the sputtering mode for the epitaxial film and predict its behavior under the external factors' influence.

A description of the substrate surface potential is necessary for modeling the growth of the epitaxial film. The film deposition model on the substrate [1] requires solving the equations of motion for an atom near the surface. A system of equations based on two-particle potentials is solved to simulate the epitaxial film growth in this model. The surface potential model greatly facilitates this task, since it requires a description of the motion of one particle in the outer field. Additionally, the surface potential is necessary when describing the scattering of particles on the surface [2]. Information on the position of maxima and minima for surface potential plays a significant role in this model.

Elastic two-dimensional atom chains are used to model thin films. There are two main approaches for describing the interaction of such a chain with a substrate. The first approach is based on the periodic potential of the Frenkel–Kontorova model [3]. The basic Frenkel–Kontrova model [3] studies a one-dimensional elastic atoms chain in the substrate potential. The cosine function specifies the periodic substrate potential. This simple approximation reveals a wide variety of ground states for the system [4–6] and different waves in the atoms chain [7–9]. Solitons are of the greatest interest. Calculations for the two-dimensional case

show ordered structures in a monoatomic film [10–12]. These structures alter the magnetic properties of the ferromagnetic film on the substrate. There are several generalizations of the Frenkel–Kontorova model. The potentials of these models contain wide maximums and minimums [13] and additional local minimums [14].

The second approach considers the paired interactions between atoms. The potential energy for the interaction of the substrate with the film atoms is equal to the sum of the substrate atom's potential. Paired interactions use analytic functions in most cases. Some models use discrete table values. The analytical functions of pairwise interaction contain parameters. The selection of these parameters is based on empirical data. Potential parameters vary to best reproduce experimental results. Such models require great computational power to obtain a surface potential landscape. Therefore, this approach is used to calculate the interaction of individual atoms with a substrate in the simulation of adsorption [15–18] or to calculate the shape and size of nanoclusters on a substrate [19–24].

Two-particle potentials include terms for repulsion when atoms approach and terms for the attraction when atoms are removed. The two-particle Lennard-Jones potential [25] is the most common. Parameter values for different chemical elements are empirically derived for this potential [26]. Morse potential uses quantum mechanical computation to describe atomic interaction [27]. There are several model potentials that consider chemical bonds: Tersoff potential [28], EDIP (Environment-Dependent Interatomic Potential) potential [29,30], Brenner potential [31], ReaxFF potential [32]. Special model potentials have been proposed for some chemical elements. These potentials approximate the experimental relationship between interaction and distance more precisely. The Finnis–Sinclair potential [33] simulates the interaction of metal atoms. The parameters of this potential are calculated for most metals. Todd and Linden-Bell's potential [34] is based on Sutton–Chen formalism. It also simulates the interaction between metal atoms. The first principles are also used to calculate interaction parameters for atoms of certain materials [35,36].

Two-particle potentials are mainly used to model the thermal and mechanical properties of pure substances. The description of the interaction between two different substances has two problems. The first problem is that it is difficult to determine the potential parameters for atoms of two different chemical elements. There are different schemes for calculating these parameters. These schemes use parameters for pure substances [26]. The second problem is the need to use large computing resources for calculation.

The Frenkel–Kontorova model derives general results for the behavior of epitaxial films on a substrate. The application of these results to specific substances is challenging. Models with two-part potential obtain results for specific coating substrate substances. The surface potential of the substrate is modeled in this paper for an atoms array with a square lattice on the surface. The simulation uses Lennard-Jones's two-particle potential. The results are compared with the Frenkel–Kontorova model. The effect of the Frenkel–Contour potential corrections on the monoatomic film's ground state is calculated.

## 2. Substrate Potential Modeling

The substrate is modeled as a two-dimensional lattice with fixed atoms. The substrate atoms are located in the nodes of the square lattice. The one-dimensional atoms chain is considered in the first step. Square lattice symmetry is used to generalize the results to a two-dimensional case.

We use the two-particle Lennard-Jones potential [25] to interact with substrate and film molecules. This potential satisfactorily describes the interaction between spherical non-polar molecules. The Lennard-Jones $U_{LJ}(r)$ potential is written as two terms.

$$U_{LJ}(r) = 4\varepsilon \left( \left(\frac{\sigma}{r}\right)^{12} - \left(\frac{\sigma}{r}\right)^{6} \right) \tag{1}$$

$r$ is the distance between the centers of the particles. $\varepsilon$ is the depth of the potential hole. $\sigma$ is the distance at which the interaction energy is zero. The minimum potential is located

at the distance $r_0 = \sigma \sqrt[12]{2}$. Potential describes particle repulsion at $0 < r < \sigma$. Molecules are attracted due to dipole–dipole interaction (van der Waals force) at $r > \sigma$.

We consider an infinite linear atoms chain with an interatomic distance $b$. The chain is located along the axis $OX$ (Figure 1). The potential is calculated at a point located at a distance $z$ from the atom's chain and at a distance $x$ along the $OX$ axis from the atom number $n = 0$.

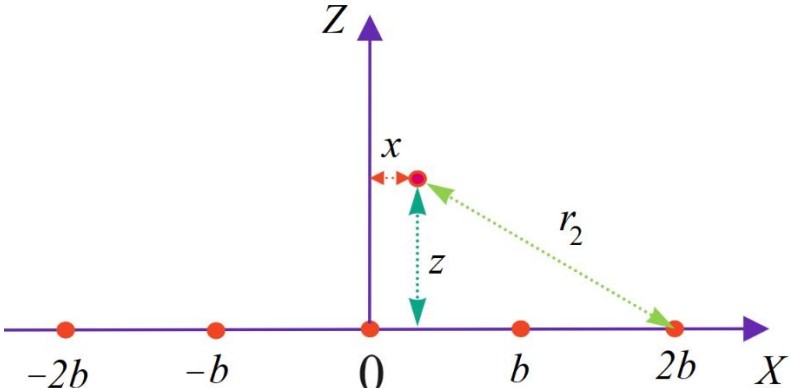

**Figure 1.** Systems geometry.

The potential at $(x, z)$ is the sum of the potentials created by all atoms in the chain.

$$U_0(x, z) = \sum_{n=-\infty}^{+\infty} U_{LJ}(r_n) \tag{2}$$

$r_n$ this is the distance from the point $(x, z)$ to the atom of the chain with the number $n$. The potential at $(x, z)$ is written based on the Lennard-Jones Formula (1).

$$U_0(x, z) = 4\varepsilon \sum_{n=-\infty}^{+\infty} \left( \left( \frac{\sigma}{r_n} \right)^{12} - \left( \frac{\sigma}{r_n} \right)^{6} \right) = 4\varepsilon \sum_{i=-\infty}^{+\infty} \left( \left( \frac{\sigma^2}{r_n^2} \right)^{6} - \left( \frac{\sigma^2}{r_n^2} \right)^{3} \right) \tag{3}$$

The formula for the distance is substituted into the potential $U_0(x, z)$.

$$U_0(x, z) = 4\varepsilon \sum_{n=-\infty}^{+\infty} \left( \left( \frac{\sigma^2}{z^2 + (nb - x)^2} \right)^{6} - \left( \frac{\sigma^2}{z^2 + (nb - x)^2} \right)^{3} \right) \tag{4}$$

Relative variables are entered for ease of calculation.

$$\sigma' = \frac{\sigma}{b}, \ x' = \frac{x}{b}, \ z' = \frac{z}{b}, U(x', z') = \frac{U_0(x, z)}{\varepsilon} \tag{5}$$

We write the potential in relative variables.

$$U(x', z') = 4 \sum_{n=-\infty}^{+\infty} \left( \left( \frac{\sigma'^2}{z'^2 + (n - x')^2} \right)^{6} - \left( \frac{\sigma'^2}{z'^2 + (n - x')^2} \right)^{3} \right) \tag{6}$$

The substrate atoms chain is periodic with period $b$. The potential $U(x', z')$ is also periodic over the variable $x'$ with period 1. Therefore, the function $U(x', z')$ is examined in the interval $0 \le x' \le 1$. The function $U(x', z')$ decomposes into a Fourier series over a variable $x'$.

$$U(x', z') = \sum_{k=0}^{\infty} A_k(z', \sigma') \cos(2\pi k x') \tag{7}$$

The coefficients $A_k(z', \sigma')$ are numerically determined. The dependence of amplitudes $A_k(z', \sigma')$ on the parameter $\sigma'$ at $z' = 1$ is shown in Figure 2. Calculations were made for a chain of $N = 201$ atoms.

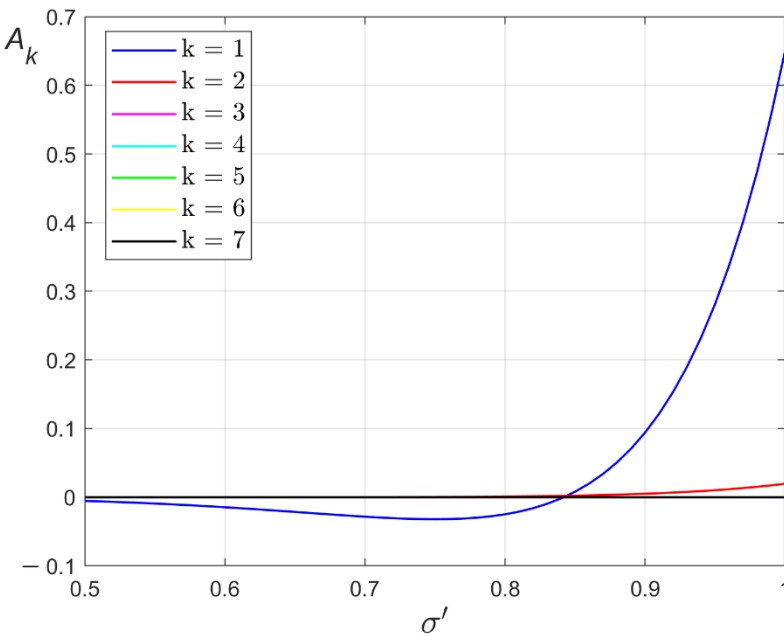

**Figure 2.** The dependence of amplitudes $A_k(z', \sigma')$ on the parameter $\sigma'$ at $z' = 1$.

The amplitude $A_1$ modulus is significantly higher than those of the remaining amplitudes $A_k$ ($k = 2, 3, \ldots$) at almost values $\sigma'$. The exception is the small area around the point $\sigma_0$ ($A_1(z', \sigma_0) = 0$). Such a point exists at all values $z'$. The position of the point $\sigma_0$ depends on $z'$. The range of possible $\sigma'$ values for different substances is calculated based on the Lennard-Jones potential parameters [37] and crystal lattice periods. This range is $0.8 \le \sigma' \le 1.2$ for metals. The first two terms dominate the rest in the Fourier series for potential (7) at $\sigma' > 0.9$. The potential is approximated by a simple formula in this case.

$$U(x', z') = \sum_{k=0}^{\infty} A_k(z', \sigma') \cos(2\pi k x') \qquad (8)$$

The difference between this potential and the exact value (7) does not exceed 1.5%. Potential (8) coincides with Frenkel–Kontorova potential up to the constant and linear transformation of coordinates $x'' = x' + 1/2$.

$$U_1(x'', z') = A_0(z', \sigma') - A_1(z', \sigma') + A_1(z', \sigma')(1 - \cos(2\pi x'')) \qquad (9)$$

The Frenkel–Kontorova model is applicable with high accuracy in $\sigma' > 0.9$.

The range of values near the $\sigma_0$ point is very interesting. Plots for $A_1(z', \sigma')$ and $A_2(z', \sigma')$ versus $\sigma'$ near $\sigma_0$ are shown in Figure 3.

As can be seen from Figure 3, the point for which $A_2(z', \sigma')$ does not match the $\sigma_0$. Therefore, there is a range of values $(z', \sigma')$ in which $A_1(z', \sigma')$ and $A_2(z', \sigma')$ are commensurate in magnitude or $A_2(z', \sigma')$ dominate $A_1(z', \sigma')$. The plot of the ratio $A_2(z', \sigma') / A_1(z', \sigma')$ coefficients to $\sigma'$ at $z' = 1$ is shown in Figure 4.

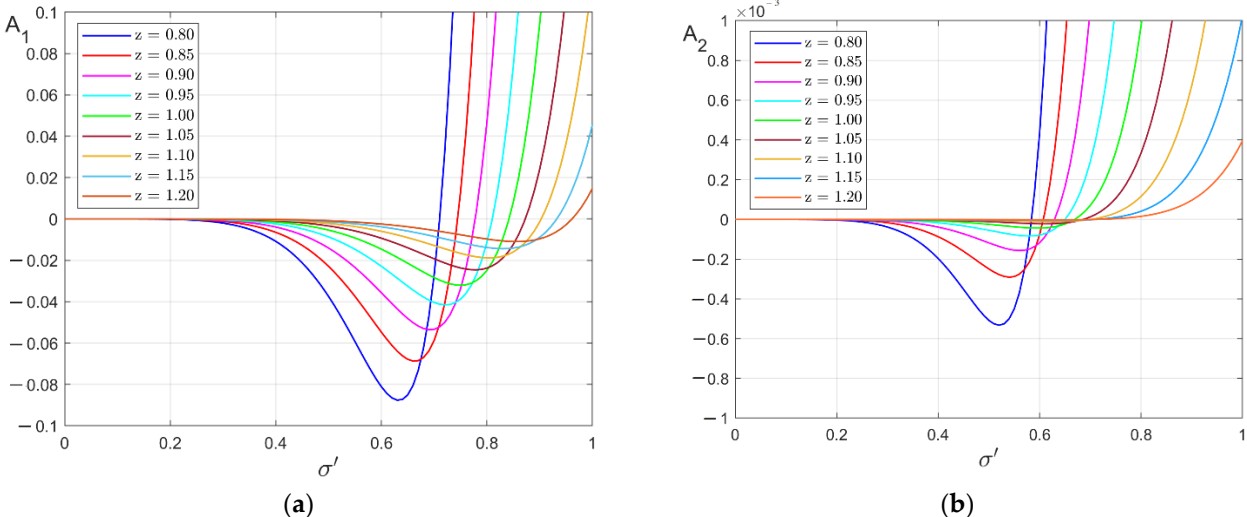

**Figure 3.** Plots for $A_1(z',\sigma')$ (**a**) and $A_2(z',\sigma')$ (**b**) versus $\sigma'$ near $\sigma_0$ $(A_1(z',\sigma_0)=0)$ at different $z'$.

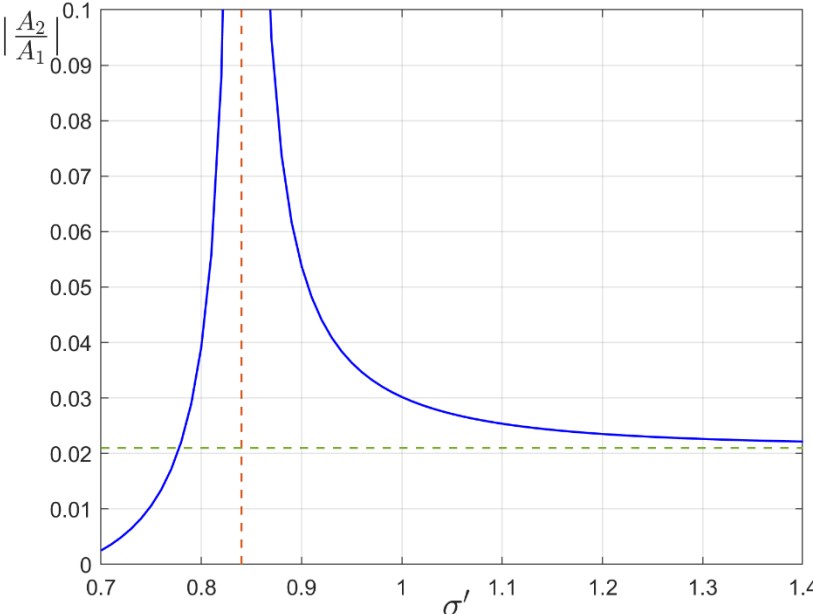

**Figure 4.** The plot of the ratio $A_2(z',\sigma')/A_1(z',\sigma')$ coefficients to $\sigma'$ at $z'=1$.

The effect of the coefficient $A_2(z',\sigma')$ on the potential value becomes more significant when approaching the point $\sigma_0$. Therefore, the question about the position of the points for which $A_1(z',\sigma')=0$ is important. The plot of the parameter $\sigma_0$ dependence on the distance to the film $z'$ is shown in Figure 5.

The value $\sigma_0$ increases monotonously as the distance from the substrate atoms increases. The potential of the atom's chain is approximated by the formula with two terms at the point $\sigma_0$.

$$U_2(x',z') = A_0(z',\sigma_0) + A_2(z',\sigma_0)\cos(4\pi x') \tag{10}$$

This expression also boils down to the Frenkel–Kontorova potential. The potential $U_2(x',z')$ has a period of two times less than that for the substrate atoms. The three terms in potential are considered in near the point $\sigma_0$.

$$U_3(x',z') = A_0(z',\sigma') + A_1(z',\sigma')\cos(2\pi x') + A_2(z',\sigma')\cos(4\pi x') \tag{11}$$

We use coordinate shift $x'' = x' + 1/2$.

$$U_3(x'', z') = A_0(z', \sigma') - A_1(z', \sigma') + A_1(z', \sigma')(1 - \cos(2\pi x'') + B(z', \sigma')\cos(4\pi x'')) \tag{12}$$

where $B(z', \sigma') = A_2(z', \sigma')/A_1(z', \sigma')$. The potential $U_3(x'', z')$ has global and local minima. The potential period $U_3(x'', z')$ coincides with the period of the substrate atoms.

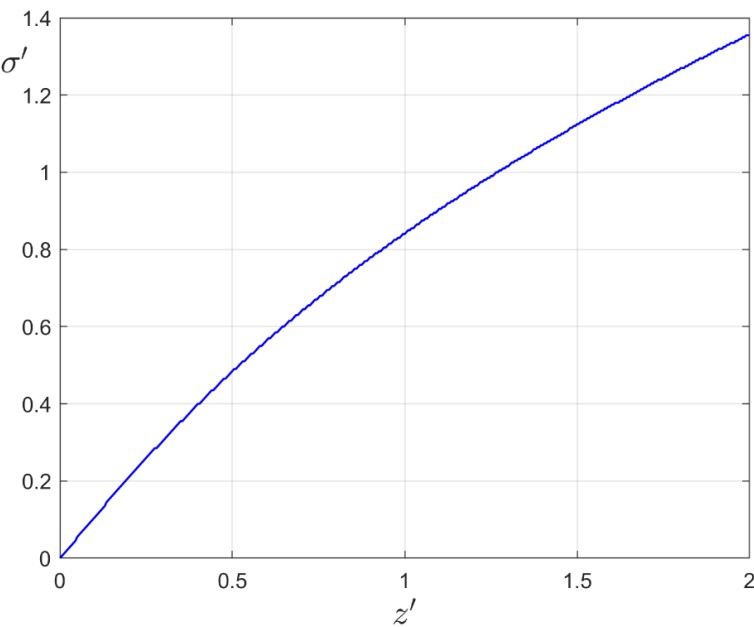

**Figure 5.** The plot of the parameter $\sigma_0$ dependence on the distance to the film $z'$.

We investigate the dependence of simulation results on the number of atoms in the chain. The dependence of the coefficient $B(1, 1)$ on the number of atoms in the chain is shown in Figure 6.

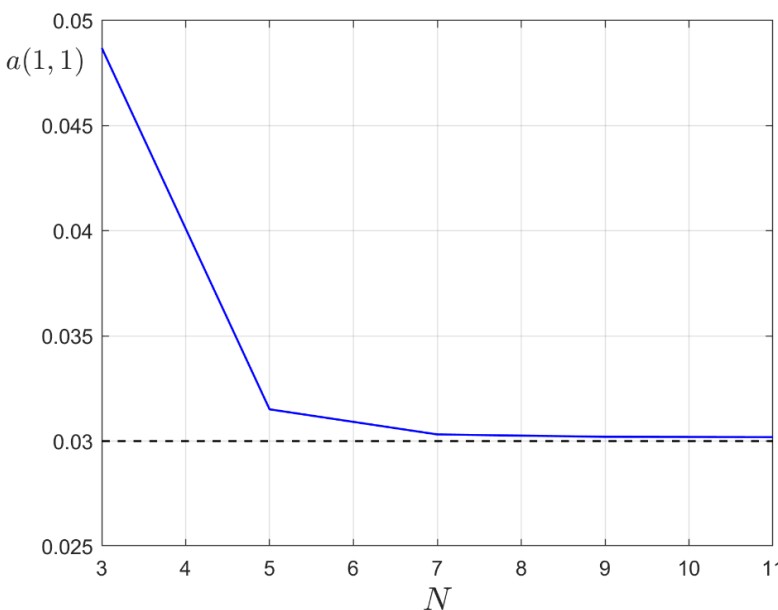

**Figure 6.** The dependence of the coefficient $B(1, 1)$ on the number of atoms in the chain.

Calculations show the invariability of the value $B(1, 1)$ with an increase in the number of atoms more than seven ($N > 7$). The same result is observed for other values $z'$ and $\sigma'$. A plot for the potential $U(x', z')$ versus $x'$ at $z' = 1$, $\sigma' = 1$, and $\varepsilon = 1$ for different numbers of atoms in the chain $N$ is shown in Figure 7.

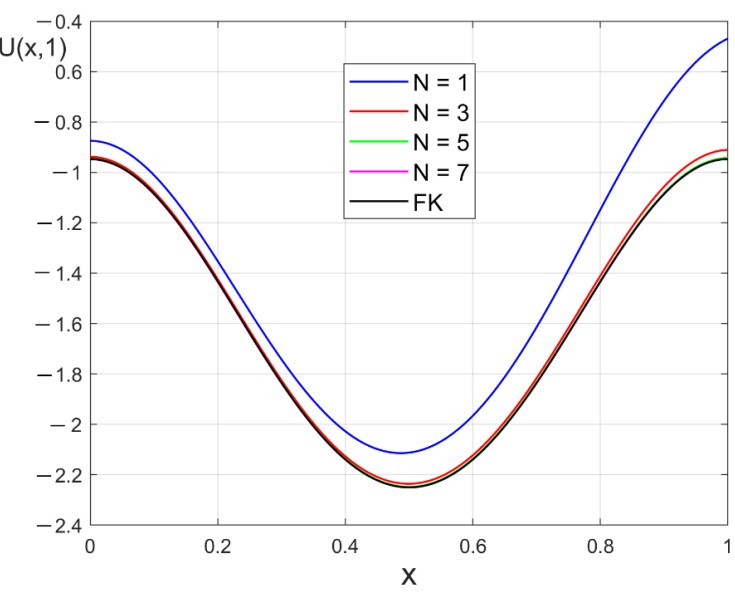

**Figure 7.** A plot for the potential $U(x', z')$ versus $x'$ at $z' = 1$, $\sigma' = 1$, and $\varepsilon = 1$ for different numbers of atoms in the chain $N$. FK is the Frenkel–Kontorova potential.

## 3. Ground State of Monoatomic Film

We consider the effect of substrate parameters on the ground state of the monolayer film on its surface. The substrate atoms form a square lattice on its surface in modeling. The substrate potential is written as a product of two factors similar to the potential $U_3$, considering the symmetry of the atom's arrangement.

$$V(x, y) = A(1 - \cos(2\pi x) + B \cos(4\pi x))(1 - \cos(2\pi y) + B \cos(4\pi y)) \tag{13}$$

The constant term is not considered in this formula. The potential energy is determined up to a constant. The substrate potential amplitude $A = \varepsilon A_1^2(z', \sigma')$ and the multiplier $B = A_2(z', \sigma') / A_1(z', \sigma')$ depend on the type of substrate atoms and film atoms. This relationship is determined by the Lennard-Jones potential parameters and lattice period $b$. The potential amplitude $A$ and the multiplier $B$ change under substrate deformations. Strains are caused by mechanical compression or stretching, heating, or electric field in the case of ferroelectrics.

The interaction between the film atoms is elastic.

$$U_{int} = \frac{g}{2} \sum_{n,m} \left( (x_{n+1,m} - x_{n,m} - a)^2 + (y_{n,m+1} - y_{n,m} - a)^2 \right) \tag{14}$$

$g$ is the modulus of elasticity, $(x_{n,m}, y_{n,m})$ are the coordinates of the film atom in the node. $a$ is the crystal lattice period for the monolayer film. The film atoms are located in the nodes of the square lattice in an undisturbed state. Distances are measured in terms of lattice period $b$. We consider the case of equality in lattice and film periods $(a/b = 1)$. The interaction energy of the film atoms is written as

$$U_{int} = \frac{g}{2} \sum_{n,m} \left( (x_{n+1,m} - x_{n,m} - 1)^2 + (y_{n,m+1} - y_{n,m} - 1)^2 \right) \tag{15}$$

The total potential energy of the film atoms in the substrate potential field is calculated as the total energy.

$$\begin{aligned} E = \frac{g}{2} \sum_{n,m} \left( (x_{n+1,m} - x_{n,m} - 1)^2 + (y_{n,m+1} - y_{n,m} - 1)^2 \right) \\ + A \sum_{n,m} (1 - \cos(2\pi x_{n,m}) + B \cos(4\pi x_{n,m}))(1 - \cos(2\pi y_{n,m}) + B \cos(4\pi y_{n,m})) \end{aligned} \tag{16}$$

The ground state minimizes total energy.

$$E \to min. \tag{17}$$

This condition determines the coordinates of atoms $(x_{n,m}, y_{n,m})$. Minimization was performed in a computer experiment. The film atoms are in the nodes of the square lattice in the initial state. The sequential iteration method is used to find the ground state. Each atom is shifted to a random vector at each iteration. If the new position of the atom reduces the energy of the system, then it is received, otherwise, the atom returns to its original state. The studies are performed for different values of parameter $B$. The arrangement of film atoms for elasticity modulus equal to substrate amplitude ($g/A = 1$) and different values of parameter $B$ are shown in Figure 8.

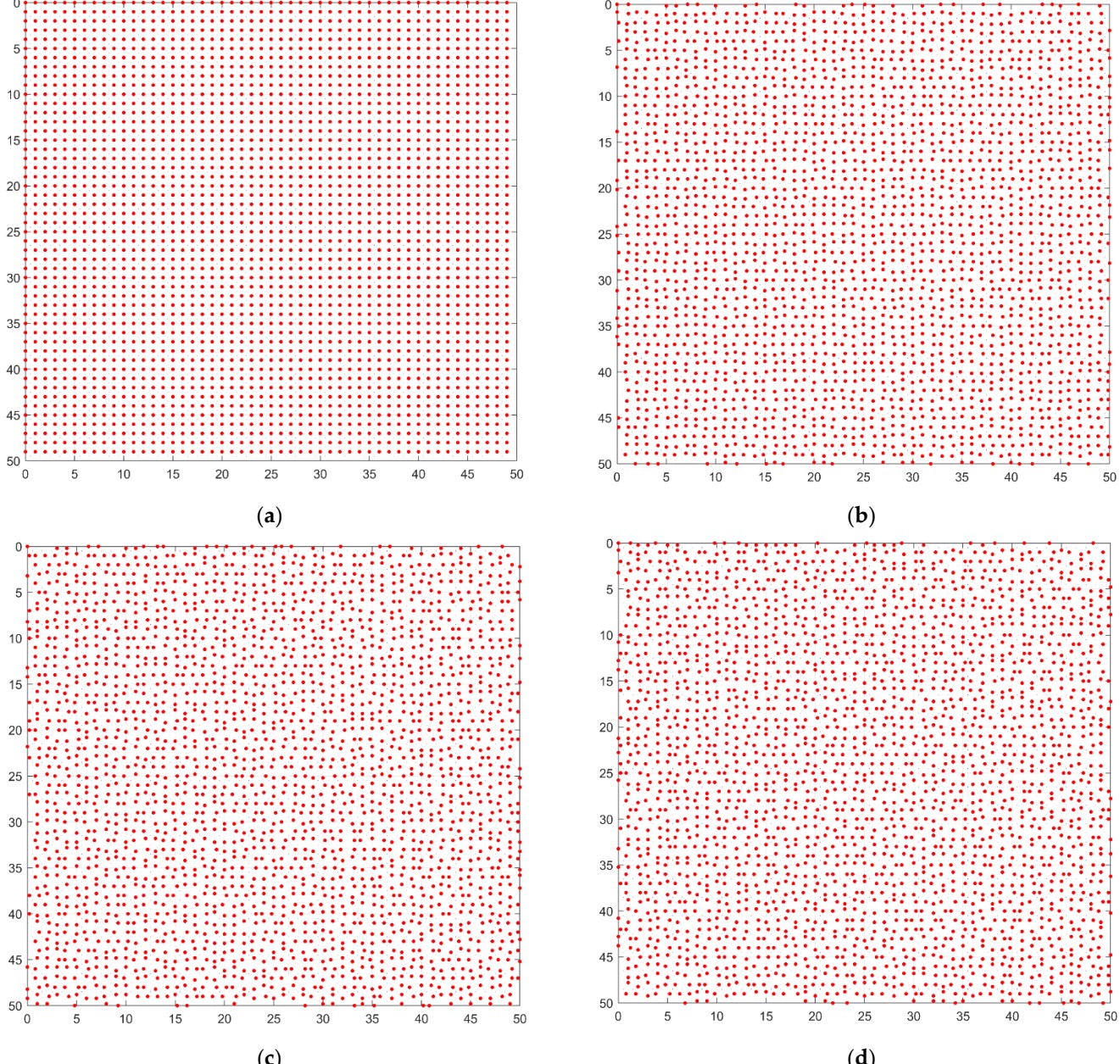

**Figure 8.** The arrangement of film atoms for elasticity modulus equal to substrate amplitude ($g/A = 1$) and different values of parameter $B$: (**a**) $B = 0.1$, (**b**) $B = 0.5$, (**c**) $B = 1.0$, (**d**) $B = 2$.

Increasing the parameter $B$ disrupts the arrangement of the film atoms. Atoms are located in the nodes of the square lattice at small values of parameter $B$. This state corresponds to the crystalline phase. The displacement of atoms occurs when the parameter $B$ increases. The film enters the amorphous phase. The far order is absent in this phase, but the near order is present. The change in the atom's location is caused by the change of a substrate potential landscape. Substrate potential surfaces for different values of parameter $B$ are shown in Figure 9.

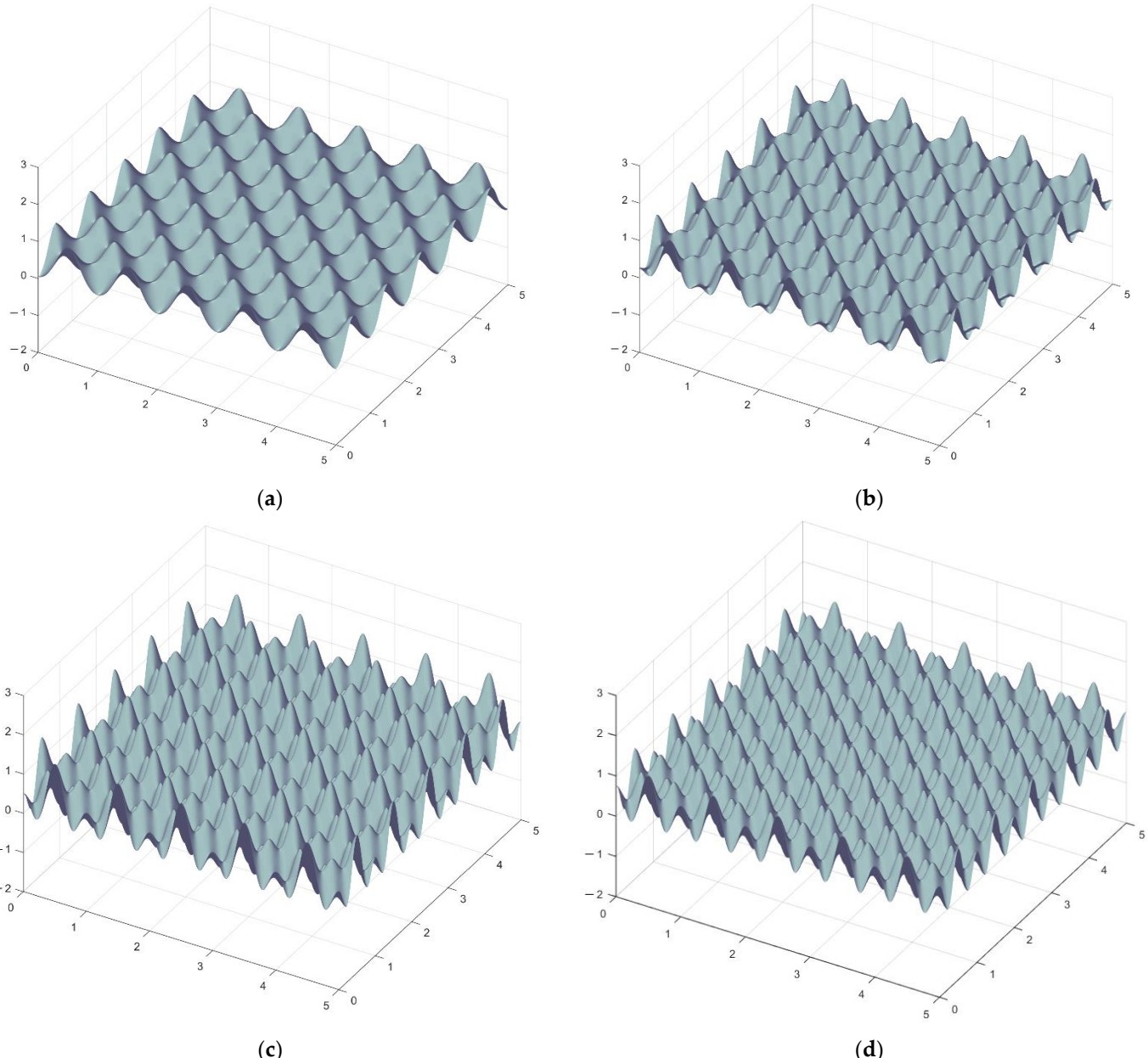

(**a**)

(**b**)

(**c**)

(**d**)

**Figure 9.** Substrate potential surfaces for different values of parameter $B$: (**a**) $B = 0.1$, (**b**) $B = 0.5$, (**c**) $B = 1.0$, (**d**) $B = 2$.

Additional local highs and lows appear in the potential landscape as parameter $B$ increases. The periodicity of the substrate potential ceases to coincide with the period of the film, which leads to the displacement of its atoms.

### 4. Structural Phase Transition

The change in the geometry of the epitaxial film atoms when the parameter $B$ changes is a structural phase transition induced by the substrate. This phase transition occurs only near the point $\sigma_0$. Certain values of the lattice period for the substrate and the distance from the substrate atoms to the film atoms are necessary to observe the phase transition.

We enter the order parameter $\varphi$ to describe the phase transition. The order parameter is one ($\varphi = 1$) in a fully ordered phase (two-dimensional crystal lattice) and is zero ($\varphi = 0$) in a completely disordered phase (chaotic arrangement of atoms). The atoms are placed in the nodes of the square lattice in a fully ordered phase. The coordinates of the atom number $(n, m)$ are determined based on the basis vectors $\vec{e}_x$, $\vec{e}_y$.

$$\vec{r}_{n,m} = n\vec{e}_x + m\vec{e}_y \tag{18}$$

The position of the atom number $(n, m)$ in the fully ordered phase is $\vec{r}_{n,m}^{(0)}$. $\langle \delta r \rangle$ this is the average deviation from the equilibrium position by one atom.

$$\langle \delta r \rangle = \lim_{N \to \infty} \frac{1}{N} \sum_{k=1}^{N} |\vec{r}_{n,m} - \vec{r}_{n,m}^{(0)}| \tag{19}$$

$N$ is the number of atoms. The mean deviation from the equilibrium position is zero $\langle \delta r \rangle = 0$ in a fully ordered phase. The atoms are randomly located in a completely disordered phase. The average density of atoms in a completely disordered phase is equal to the density of atoms in a completely ordered phase. One crystal lattice node in an ordered phase corresponds to one atom. We calculate the mean deviation $\langle \delta r \rangle_{rand}$ for a completely disordered phase. Each atom is randomly located in the region: $-a/2 < x - x_0 < a/2; -a/2 < y - y_0 < a/2$. The $\langle \delta r \rangle_{rand}$ tends to a constant value when $N \to \infty$.

$$\langle \delta r \rangle_{rand} = \frac{1}{a^2} \iint_{-a/2}^{a/2} \sqrt{x^2 + y^2} dx dy \approx 0.3826a. \tag{20}$$

The order parameter is determined based on the mean deviation for a fully ordered phase and a completely disordered phase.

$$\varphi = 1 - \frac{\langle \delta r \rangle}{\langle \delta r \rangle_{rand}} \tag{21}$$

This determination of the order parameter satisfies the requirements for its values in the ordered phase and the disordered phase. The plot for the dependence of the order parameter $\varphi$ on the parameter $B$ is shown in Figure 10.

Numerical calculations were carried out for a lattice with $200 \times 200$ atoms. The order parameter is one ($\varphi = 1$) when $B \leq 0.275$. The film is in a fully ordered phase. The order parameter jumps in the range $0.275 \leq B \leq 0.276$. The transition to the unordered phase occurs in this interval. The order parameter decreases monotonously, aiming for an asymptotic value $\varphi \approx 0.3466$ at $B \geq 0.276$. The system does not enter a completely disordered phase. There is a near order in the arrangement of atoms, but there is no far order. This state corresponds to the amorphous phase in the film.

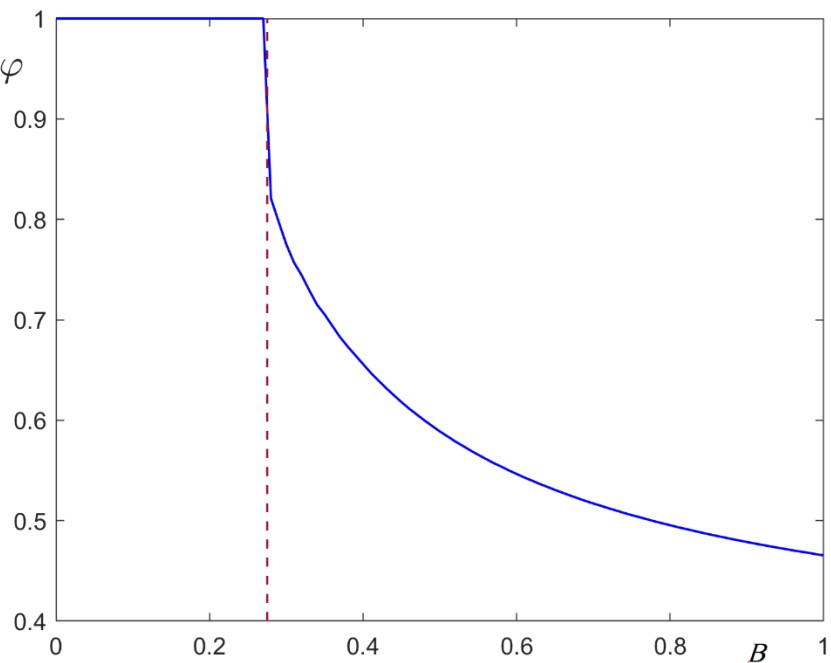

**Figure 10.** The plot for the dependence of the order parameter $\varphi$ on the parameter $B$.

## 5. Discussion

The transition from a two-particle potential to a collective potential depends on the kind of atoms and their mutual location. The potential period coincides with the substrate atom's period in most cases. The potential coincides with the Frenkel–Kontorova potential. In this case, the thin film atoms are located at the minimum potential. The film is in the crystalline phase. There is a narrow range of system parameter values near which the potential has a more complex landscape. Additional highs and lows are present in this area. Changing the substrate potential affects the thin film state. The destruction of the far order occurs in the film. The film enters the amorphous phase.

The effect of the substrate state on the crystallinity of thin films has been observed in various experimental works. Films $Ge_8Sb_{92}$ transition from the amorphous phase to the crystalline phase when the substrate is heated [37]. The crystallinity of the thermoelectric thin film Ag-Sb-Te at room temperature depends on the state of the substrate [38]. The crystalline silicon thin film is a two-phase structure of the amorphous phase and the crystalline phase [39]. The transition temperature from the amorphous phase to the crystalline phase shifts under the influence of the substrate.

The proposed model describes only pure materials for the substrate and epitaxial film. The film and substrate are multi-component systems in many real experiments. The description of such systems requires an extension of the proposed model. The multi-component system model includes several embedded crystal lattices in the description of the substrate. The difference in the elasticity coefficients for different atom pairs and the parameters of the different interactions with the substrate must be considered when describing multicomponent films. Some researchers have obtained experimental results for the microstructure and mechanical properties of pure films on the substrate [40–42]. These results show that changing the surface potential by introducing additional components increases the resistance of the coatings.

**Author Contributions:** Conceptualization, S.V.B. funding acquisition, I.V.B.; investigation, S.V.B. and I.V.T.; methodology, I.V.B. and S.V.B.; project administration, I.V.B.; software, I.V.T.; writing—original draft, S.V.B. and I.V.T. All authors have read and agreed to the published version of the manuscript.

**Funding:** This research was funded by the Russian Science Foundation, project number 22-19-00355.

**Institutional Review Board Statement:** Not applicable.

**Informed Consent Statement:** Not applicable.

**Data Availability Statement:** Not applicable.

**Conflicts of Interest:** The authors declare no conflict of interest.

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
