# Peer review of "Simulation of Epitaxial Film–Substrate Interaction Potential"

_coatings, doi:10.3390/coatings12060853_

Round 1

Reviewer 1 Report

Reviewer's comments:

The Frenkel-Kontorova model derives general results for the behavior of epitaxial films on a substrate. The application of these results to specific substances is challenging. Models with two-part potential obtain results for specific coating substrate substances. The surface potential of the substrate is modeled in this paper for an atoms array with a square lattice on the surface. The simulation uses Lennard-Jones's two-particle potential. The results are compared with the Frenkel-Kontorova model. The effect of the Frenkel Contour potential corrections on the monoatomic films ground state is calculated. Although this paper presents some new results, some important research background, research significance and experimental verification should be introduced. Therefore, some major revisions should be made before publication.

1. The author should give the information of the country and region.

2. The interaction between the epitaxial film and the substrate has an important effect on the growth and crystal structure of the coating. In the introduction, there is a lack of description of the research background and significance. It is suggested that the author should supplement the research background and significance of this work and its guiding role in surface coating technology.

3. Two-particle potentials are mainly used to model the thermal and mechanical properties of pure substances。However, in many cases, the film and substrate are often a complex multi-component system. The composition of the film is not pure material, but a variety of phases. When using two particle potentials to simulate pure substances, the author must explain the limitations of this work and the guiding significance for the preparation of thin films.

4. Two particle potential is used to simulate the thermal and mechanical properties of pure materials. Whether this simulation work can be verified by experimentally prepared pure material coatings. Some scholars have investigated the interaction of si-w/mo binary system, and investigated the microstructure and mechanical properties of pure MoSi2 and WSi2 films on Mo/W substrate, such as, Microstructure evolution and growth mechanism of Si-MoSi2 composite coatings on TZM (Mo-0.5Ti-0.1Zr-0.02 C) alloy; Microstructure and oxidation resistance of Si-MoSi2 ceramic coating on TZM (Mo-0.5Ti-0.1Zr-0.02C) alloy at 1500 ℃; Evolution of surface morphology, roughness and texture of tungsten disilicide coatings on tungsten substrate; Improving oxidation resistance of TZM alloy by deposited Si–MoSi2 composite coating with high silicon concentration. It is suggested that the authors quote their reports in order to highlight the importance of the simulation work.

5. What are the effects of substrate potential surfaces and film atoms arrangement on the growth and structural phase transformation of thin films?

6. Can this theory be used to explain the growth and structural transformation of pure MoSi2 and WSi2 coatings on Mo/W substrates (Synthesis WSi2 coating on W substrate by HDS method with various deposition times; Formation of MoSi2 and Si/MoSi2 coatings on TZM (Mo–0.5Ti–0.1Zr–0.02C) alloy by hot dip silicon-plating method; Oxidation protection of tungsten alloys for nuclear fusion applications: A comprehensive review)? I suggested that the author refer to these reports to verify your simulation conclusions, which may arouse the wide interest of more researchers.

Author Response

Уважаемый рецензент!

We thank you for your constructive comments. We provide answers to your questions and comments.

  1. The author should give the information of the country and region.

Answer: Country and region information added.

  1. The interaction between the epitaxial film and the substrate has an important effect on the growth and crystal structure of the coating. In the introduction, there is a lack of description of the research background and significance. It is suggested that the author should supplement the research background and significance of this work and its guiding role in surface coating technology.

Answer:

A paragraph has been added in the introduction.

“The substrate potential determines the basic properties of epitaxial films. The films crystal lattice is formed in the sputtering step under the influence of the substrate. The relative arrangement of the film atoms is determined by the substrate potential. The mechanical, thermal, magnetic and optical properties of the epitaxial film can be altered by acting on the substrate. Changing the size or symmetry of a substrate crystal lattice changes its surface potential. The calculation of the substrate surface potential makes it possible to select the sputtering mode for the epitaxial film and predict its behavior under the external factors influence.”

  1. Two-particle potentials are mainly used to model the thermal and mechanical properties of pure substances。However, in many cases, the film and substrate are often a complex multi-component system. The composition of the film is not pure material, but a variety of phases. When using two particle potentials to simulate pure substances, the author must explain the limitations of this work and the guiding significance for the preparation of thin films.

Answer:

A paragraph has been added in the Discussion.

“The proposed model describes only pure materials for the substrate and epitaxial film. The film and substrate are multi-component systems in many real experiments. The description of such systems requires an extension of the proposed model. The multi-component system model includes several embedded crystal lattices in the description of the substrate. The difference in the elasticity coefficients for different atoms pairs and the different interactions parameters with the substrate must be considered when describing multicomponent films.”

  1. Two particle potential is used to simulate the thermal and mechanical properties of pure materials. Whether this simulation work can be verified by experimentally prepared pure material coatings. Some scholars have investigated the interaction of si-w/mo binary system, and investigated the microstructure and mechanical properties of pure MoSi2 and WSi2 films on Mo/W substrate, such as, Microstructure evolution and growth mechanism of Si-MoSi2 composite coatings on TZM (Mo-0.5Ti-0.1Zr-0.02 C) alloy; Microstructure and oxidation resistance of Si-MoSi2 ceramic coating on TZM (Mo-0.5Ti-0.1Zr-0.02C) alloy at 1500 ℃; Evolution of surface morphology, roughness and texture of tungsten disilicide coatings on tungsten substrate; Improving oxidation resistance of TZM alloy by deposited Si–MoSi2 composite coating with high silicon concentration. It is suggested that the authors quote their reports in order to highlight the importance of the simulation work.

Answer:

A general approach to modeling the surface substrate potential and its effect on epitaxial films is proposed in this paper. We are currently working to refine this model for specific materials. We hope to present our results for some materials soon. But these results are quite voluminous and constitute the content of a separate article. However, we have added the following text to the “Discussion” le and additional links to publications.

“Some researchers have obtained experimental results for the microstructure and mechanical properties of pure films on the substrate [40-42]. These results show that changing the surface potential by introducing additional components increases the resistance of the coatings.”

  1. Как влияют потенциальные поверхности подложки и расположение атомов пленки на рост и структурно-фазовое превращение тонких пленок?

Ответ :

Во введении добавлен абзац.

«Описание поверхностного потенциала подложки необходимо при моделировании роста эпитаксиальных пленок. Модель осаждения пленки на подложку [1] требует решения уравнений движения атома вблизи поверхности. Для моделирования эпитаксиального роста пленки в этой модели решается система уравнений, основанная на двухчастичных потенциалах. Модель поверхностного потенциала значительно облегчает эту задачу, так как требует описания движения одной частицы во внешнем поле. Также поверхностный потенциал необходим при описании рассеяния частиц на поверхности. [2]. Информация о положении максимумов и минимумов поверхностного потенциала играет важную роль в этой модели».

  1. Можно ли использовать эту теорию для объяснения роста и структурной трансформации чистых покрытий MoSi2 и WSi2 на подложках Mo/W (Синтез покрытия WSi2 на подложке W методом ГДС с различным временем осаждения; Формирование покрытий MoSi2 и Si/MoSi2 на TZM (Mo –0,5Ti–0,1Zr–0,02C) методом горячего кремниевого покрытия погружением; Защита от окисления вольфрамовых сплавов для применения в ядерном синтезе: всесторонний обзор)? Я предложил автору обратиться к этим отчетам, чтобы проверить ваши выводы о моделировании, которые могут вызвать широкий интерес у большего числа исследователей.

Ответ :

Как я указал в ответе на один из предыдущих комментариев, сейчас мы разрабатываем нашу модель для описания конкретных систем. Это исследование еще не завершено, но предварительные результаты обнадеживают. Мы планируем опубликовать новые результаты в ближайшее время. Система, которую вы описываете, также будет рассмотрена.

Reviewer 2 Report

The paper studies crystalline ordering using a point interaction derived from Lenard Jones type two body potentials. It is largely sound and interesting. I have some clarification questions:

1. In the calculation of the structural phase transition the order parameter going to zero for amorphous phases is not well defined. In particular, what is meant by a completely random ordering ?

2. For the numerical simulation update does each run update the positions of all the particles ? Why is the procedure adopted a one shot accept, why not a probability for accepting the moves depending on some cutoff scale ?

3.  It is very difficult to understand the nature of the transitions as B is increased in the simulations. It would be better to track the mean distance or some other factor.

Author Response

Dear reviewer!

We thank you for your constructive comments. We provide answers to your questions and comments.

  1. In the calculation of the structural phase transition the order parameter going to zero for amorphous phases is not well defined. In particular, what is meant by a completely random ordering?

Answer:

Хаотическое распределение атомов на поверхности трактуется как совершенно случайный порядок. In this case, the coordinates of the atoms are random variables with a uniform distribution. The order parameter is zero for such a distribution of atoms. A completely random distribution corresponds to the gaseous state of the substance.

  1. For the numerical simulation update does each run update the positions of all the particles? Why is the procedure adopted a one shot accept, why not a probability for accepting the moves depending on some cutoff scale?

Answer:

The position of all film atoms is updated on each run. Calculation of probabilities is necessary in the study of film heating. In this paper, we consider substrate deformations caused by mechanical causes or ferroelectric effects. In the future, we plan to develop the model considering thermal effects.

  1. It is very difficult to understand the nature of the transitions as B is increased in the simulations. It would be better to track the mean distance or some other factor.

Answer:

The order parameter we enter is based on the displacement of the film atoms. The change in parameter B is due to the change in geometric parameters for the substrate crystal lattice. Changing lattice constant b results in changing sigma’ and z’ parameters, which change parameter B and change the surface potential landscape. The film atoms in this potential also experience displacement. In some cases, this bias is so large that it leads to a phase transition.

Round 2

Reviewer 1 Report

The present modification is reasonable and acceptable.